# Epidemiology and Ecology of Usutu Virus Infection and Its Global Risk Distribution

**DOI:** 10.3390/v16101606

**Published:** 2024-10-12

**Authors:** Jiahao Chen, Yuanyuan Zhang, Xiaoai Zhang, Meiqi Zhang, Xiaohong Yin, Lei Zhang, Cong Peng, Bokang Fu, Liqun Fang, Wei Liu

**Affiliations:** 1State Key Laboratory of Pathogen and Biosecurity, Academy of Military Medical Science, 20 Dong-Da Street, Fengtai District, Beijing 100071, China; chenjh1208@126.com (J.C.); babylovehopi@163.com (X.Z.); ic577703@163.com (M.Z.); yinxiaohong_daisy@163.com (X.Y.); zl632892@126.com (L.Z.); 13087737639@163.com (C.P.); 2Department of Disease Control and Prevention, No. 926 Hospital of Joint Logistics Support Force, Kaiyuan 661600, China; 19806934373@163.com; 3School of Public Health, Anhui Medical University, Hefei 230022, China; fbk9938@163.com

**Keywords:** Usutu virus, distribution, machine learning, ensemble learning

## Abstract

Usutu virus (USUV) is an emerging mosquito-transmitted flavivirus with increasing incidence of human infection and geographic expansion, thus posing a potential threat to public health. In this study, we established a comprehensive spatiotemporal database encompassing USUV infections in vectors, animals, and humans worldwide by an extensive literature search. Based on this database, we characterized the geographic distribution and epidemiological features of USUV infections. By employing boosted regression tree (BRT) models, we projected the distributions of three main vectors (*Culex pipiens*, *Aedes albopictus*, and *Culiseta longiareolata*) and three main hosts (*Turdus merula*, *Passer domesticus*, and *Ardea cinerea*) to obtain the mosquito index and bird index. These indices were further incorporated as predictors into the USUV infection models. Through an ensemble learning model, we achieved a decent model performance, with an area under the curve (AUC) of 0.992. The mosquito index contributed significantly, with relative contributions estimated at 25.51%. Our estimations revealed a potential exposure area for USUV spanning 1.80 million km^2^ globally with approximately 1.04 billion people at risk. This can guide future surveillance efforts for USUV infections, especially for countries located within high-risk areas and those that have not yet conducted surveillance activities.

## 1. Introduction

Usutu virus (USUV) is an emerging arbovirus classified as a member of the *Orthoflavivirus* genus in the Flaviviridae family [1,2]. It belongs to the antigenic complex of Japanese encephalitis virus (JEV) and shares close genetic relatedness with JEV and West Nile virus (WNV) [3]. Similar to WNV, USUV has an enzootic cycle involving mosquitoes as vectors and birds as amplifying hosts [4], while humans [5,6] and other mammals [7,8] can be infected through mosquito bites, but act as “dead-end” hosts.

USUV, which originated in Africa, was initially isolated from *Culex neavei* around the Usutu River in Swaziland in 1959 [9]. Subsequently, it has been reported across various regions across Africa [10]. Recently, USUV has gained significant attention due to its emergence and expansion in Europe, as well as highly diversified circulating lineages. The first report of USUV in Europe was identified in Austria in 2001 [11], followed by reports in Hungary [12], Spain [13], Switzerland [14], Germany [15], and Greece [16]. In fact, the occurrence of USUV in Europe can be traced back to Italy in 1996, when it caused a considerable number of *Turdus merula* deaths [17] but remained silent for many years. Furthermore, there have been reports of USUV spreading beyond Europe and Africa. For instance, it has been found in mosquitoes [18] and horses in Israel [19].

The expanding geographic range of USUV infections has led to increased human exposure and subsequent human infections. The first confirmed case of human USUV infection was reported in the Central African Republic in 1981, followed by a second case in Burkina Faso in 2004 [10]. In Europe, the first documented case of USUV infection in humans was reported in Italy in 2009 [6], with numerous additional cases subsequently identified [20]. While most cases are asymptomatic or present with mild unspecific symptoms such as fever, rash, headache, and others [6,21,22,23,24], some individuals might experience severe neurological complications, including meningitis or meningoencephalitis. These complications can occur not only among immunocompromised patients [5,6] but also occasionally in healthy individuals. It is noteworthy that USUV-related complications have recently shown a steady increase across European countries, including Austria [22], Croatia [25], the Czech Republic [26], and Hungary [27], with the first fatal cases being reported in Italy during 2022 [28]. Furthermore, molecular and serologic surveillance conducted across several European countries has identified USUV infections among blood donors as well [29]. With the increasing global attention on USUV infection and advancements in molecular diagnostic technology, there has been a surge in research focusing on local surveillance of USUV infection as well as modeling analysis [30,31]. However, current research efforts are limited to specific countries or regions, leaving the worldwide distribution of USUV infections in vectors and hosts and associated risk burden uncertain.

To comprehensively analyze the diversity, distribution, and potential risk associated with USUV infection, we conducted a systematic literature review on the global occurrence of USUV infections in vectors, animals, and humans. Based on this, we developed machine learning models to assess the environmental suitability of USUV and estimate the global risk distribution associated with USUV infections.

## 2. Materials and Methods

### 2.1. Literature Search and Database Establishment

We conducted a systematic literature search on PubMed, Web of Science, medRvix, and bioRvix to identify relevant articles reporting the detection of USUV. We used the specific search terms “Usutu” OR “Usutu virus” OR “USUV” to retrieve all articles reported from 1964 to 2023, without any language restrictions. After removing duplicate studies, two researchers (J-HC and Y-YZ) independently screened all identified studies based on titles or abstracts, followed by a thorough review of the full text. Studies that met any of the following criteria were excluded: (I) lacked laboratory-confirmed evidence; (II) focused on molecular or cellular structures, functions, or experimental infection or transstadial transmission; (III) were reviews; (IV) failed to provide access to the full text; and (V) drug or vaccine trials. Additionally, we reviewed the references of the included studies in order to identify any additional relevant research missed during the initial search. Any conflicts or uncertainties were resolved through negotiation by J-HC and Y-YZ or with the involvement of a third author (L-QF).

Confirmed USUV infections were determined through molecular assays or successful isolation of the virus. Positive results in human serological studies required either a fourfold increase in titer or a seroconversion of specific antibodies between the acute and convalescence phases. Virus neutralization tests (VNTs) were used to confirm positive results due to immune cross-reactivity with other *Orthoflavivirus* genus viruses, particularly WNV [20,32] (Appendix A). Data from included studies were extracted, including title, authorship, publication year, study period, sampling location, tested vector/animal species, demographical features, and so on (Appendix A). To conduct a phylogeographic analysis of USUV, we retrieved genome records of USUV from GenBank.

### 2.2. Geo-Referencing and Mapping the USUV Infections

The locations of USUV infections were categorized as either point or polygon data based on the availability of coordinates for sampling locations in the extracted data. The specific locations for polygon data were mapped using the geographic center coordinates obtained from the global administrative division map through ArcGIS Desktop 10.7. The USUV infections in mosquitoes and wildlife were mapped at the genus and order levels, respectively.

### 2.3. Data on Other Covariates

Based on their potential association with USUV infections (Appendix A), we selected 44 potential covariates as predictors for the ecological model of USUV occurrences. These covariates encompassed 15 environmental, 19 ecoclimatic, 5 biological, and 5 socioeconomic variables. To standardize the spatial resolution of these variables, we created a global grid map with a resolution of 10 km × 10 km and computed the average values of these variables for each cell within the grid (Appendix A). The sources of data used, along with their original spatial resolutions, and time ranges are provided in Appendix A.

### 2.4. Modeling Analysis USUV Infections

Although we aimed to explore the global risk distribution of USUV infections, our literature search and data extraction on a global scale revealed that USUV infection has only been reported in Europe, Africa, and some regions of the Middle East. Therefore, we hypothesize that USUV has not yet established a complete transmission chain in other regions where both its arthropod vectors and vertebrate reservoirs exist, such as North America. To present the current areas with potential risk, our modeling area was limited to Europe, Africa, and the Middle East. Before modeling the environmental suitability of USUV infections, we used a boosted regression tree (BRT) model to predict the occurrence probabilities of the primary vectors and hosts. The main vector species and host species of USUV currently lack a systematic consensus in various studies. Therefore, we established a USUV spatial distribution database by collecting data from all relevant literature to determine the primary vector and host species. In this database, only molecular detection was available for the positive infection rate in vectors. Molecular detection and serological detection were used for hosts. The total sample size and positive infection rate of molecular and serological detections in hosts were different, so we determined the primary and secondary vectors by the positive infection rate and determined the primary and secondary hosts by calculating “weighted importance.” In specific, based on a positive detection rate exceeding 0.50%, we identified three mosquito species (*Culex pipiens*, *Aedes albopictus*, and *Culiseta longiareolata*) as the primary vectors (Appendix A). By calculating “weighted importance” (Appendix A), we determined three bird hosts (*Turdus merula*, *Passer domesticus*, and *Ardea cinerea*) (Appendix A) as the primary hosts. Then, occurrence data for these three mosquito species and three bird hosts were obtained and mapped onto the grid map with a resolution of 10 km × 10 km. Briefly, data queries were performed on the Global Biodiversity Information Facility (GBIF) and VectorMap databases to acquire occurrence data for these three mosquito and hosts in Europe, Africa, and the Middle East from 2001 to 2023. The potential distributions of *Aedes albopictus* and *Culex pipiens* have been previously investigated [33,34]. However, considering the updated data since 2014 and 2022 and the different regions involved in our study compared to previous studies, as well the incorporation of additional variables in our models, we supplemented the occurrence data in the original dataset and used a new model. Additionally, relevant papers were systematically searched using “*Culex pipiens*” (the most important vector) in PubMed, Web of Science, medRvix, and bioRvix databases to identify further information sources (Appendix A). Distribution information reported in these articles was extracted by J-HC and M-QZ. Only occurrence data with specific geographic coordinates were included. Fourthly, we classified the occurrence cells of each species as “cases” while obtaining “controls” through different random sampling ratios (1:1, 1:3, and 1:5) within a spatial distance of 30 to 3000 km away from the occurrence cells. During this sampling process, the reciprocals of the normalized number of sequences for each species in GenBank were used to address survey bias caused by imbalance detection efforts between countries (Appendix A). Subsequently, a BRT model was utilized (Appendix A) and 100 rounds of fitting were performed while incorporating environmental and ecoclimatic variables to predict occurrence probabilities for each species. Furthermore, variable selection was performed based on pairwise correlations to mitigate multicollinearity among variables (Appendix A). The optimal model was determined by comparing the kappa coefficient and F1 score. The predicted average probability of each cell from this optimal model was designated the habitat suitability index (HSI). Additionally, we internally validated the HSIs of primary vectors and animal hosts by calculating the relative uncertainty for each cell (which was computed as the ratio of the 95% uncertainty intervals to the predicted HSI).

Based on the HSI of each species, we derived two new variables, the mosquito index and bird index. These variables represented the maximum HSI values among three mosquito and three bird species within the same cells, respectively. Finally, we integrated both indices with other environmental, ecoclimatic, biological, and socioeconomic covariates to model the ecological niche of USUV. We used occurrence data with specific geographic coordinates to model the occurrence probabilities of USUV, and we further constructed two comparison models using occurrence data with specific geographic coordinates and polygon occurrence records without geocoordinates within the area thresholds of 400 km^2^ and 900 km^2^, respectively. The polygon occurrence records exceeding these thresholds were excluded from modeling of both comparisons [35].

Apart from employing a BRT model, we utilized random forest (RF) (Appendix A) and least absolute shrinkage and selection operator (LASSO) logistic regression models (Appendix A) to predict the occurrence probabilities of USUV. We compared the performance of these different sampling methods between random cells and background cells located at occurrences of known USUV vector mosquitoes. Control cells were sampled at different ratios from areas located 30–3000 km away from the occurrence cells of USUV infections. The mean relative contributions (RCs) of each variable were calculated, and the best model was selected by comparing the area under curve (AUC) for each model.

After evaluating the top-performing models among BRT, RF, and LASSO, we constructed a composite learner using bootstrap aggregation (bagging) method [36]. A new model was generated by assigning weights to the prediction outcomes based on the AUC of each individual model (Appendix A). Finally, based on the predicted occurrence probabilities of USUV infections, all cells were categorized into three different risk levels: areas with no risk (below 0.256, which corresponds to the highest Youden index value in the ROC curves obtained from the test dataset in the ensemble learning models), areas with low to medium risk (0.256–0.750), and areas with high risk (above 0.750).

## 3. Results

### 3.1. Database Assembly

The search identified a total of 218 unique studies after exclusion of 635 studies that did not meet the criteria and 459 duplicates. Additionally, we included 7 studies from reference lists and 52 studies from GenBank, resulting in a total of 277 included studies (225 are listed in Appendix A). Of these, human infections were reported by 42 studies, vector infections by 59 studies, and animal infections by 146 studies (Figure 1).

### 3.2. USUV Infections in Humans

A total of 235 individuals with confirmed USUV infection were reported, with the majority concentrated in Europe and only 22 individuals reported in Africa. Italy accounted for the highest number of human infections (*n* = 141, 60%), followed by its neighboring country Austria (*n* = 27). Among all reported cases, a small proportion (16%, 37/235) occurred before 2011, with the predominant part (35/37) reported from Italy. Among cases with available information, males constituted 67% (67/100) of the total infections, with more than half being under 40 years old (56%, 32/57). Moreover, a majority of the infections (85%, 70/82) occurred between June and September. Among the 46 confirmed virus strains isolated from humans, 38 (83%) were confirmed as “Europe 2,” which was more frequently observed in Austria (24 vs. 14, which was the sum of all other countries, *p* = 0.003) (Figure 2). Co-infection of USUV with other viruses was reported among 39% (48/122) of individuals (45 with WNV and three with TOSV), predominantly observed in Italy. Clinical symptoms were obtained from 135 human infections, and a majority were asymptomatic (70–71%). Among patients with symptomatic infection, encephalitis/meningitis was most commonly recorded (20–21%), followed by fever (13–23%) and headache (10–21%). Details for each country are given in Table 1 and Appendix A. The USUV-infected individuals were primarily distributed in southern and eastern Europe, particularly Italy and Austria. However, the diagnosis methods differed, with serological testing more commonly applied in Italy, while molecular testing is more commonly applied in Austria (Figure 3A).

### 3.3. USUV Infections in Vectors

A more extensive distribution range was observed for USUV infections in vectors, spanning from 26° S to 52° N (Figure 3B). Infections in mosquitoes were primarily observed in Europe, particularly in northeastern Italy. The overall positive rate of USUV detected by PCR tests was estimated as 1.60% in mosquitoes when considering only studies that provided a clear pool number and had a total pool greater than or equal to 10 (Appendix A). In total, USUV was detected in 20 mosquito species, including six species of Aedes, seven species of Culex, four species of Anopheles, etc. (Figure 4A). Culex mosquitoes exhibited the highest positive rate at 1.92% (980/51,167), followed by Culiseta (0.77%), Aedes (0.37%), and Anopheles (0.20%). At the species level, significant variations were observed, with *Culex pipiens* showing the highest positive rate (2.28%), followed by *Culiseta longiareolata* (2.22%) and *Aedes albopictus* (0.97%), higher than the remaining 17 species (<0.50%).

### 3.4. USUV Infections in Animals

A total of nine livestock species tested positive for USUV, with an overall seroprevalence of 8.53% (1417/16,620) when considering only studies that provided a clear number and had a total sample size greater than or equal to 10. Among the positive livestock species, horses displayed the highest positive rate (9.51%, 1310/13,772), followed by chickens (7.73%) and pigs (4.65%), significantly higher than the others (detailed in Appendix A). Positive detection of USUV infections in livestock was mainly reported in Europe and Africa, with higher rates reported in Europe, particularly countries located on the northern coast of the Mediterranean, such as France and Italy (Figure 3C).

A total of 166 wildlife species belonging to 31 orders tested positive for USUV (Figure 4B), with an overall seroprevalence of 4.94% (579/11,712) (Appendix A). Among these, birds emerged as key hosts, with 146 bird species positive for USUV, yielding a positive rate of 5.26% (168/3191). The highest number of reported infected species (53 species) was observed for Passeriformes birds. Additionally, seroprevalence among birds was found to be higher than that of mammals (3.81%). USUV infections in wildlife were predominantly distributed throughout Europe, closely resembling those observed in livestock, concentrated within northeastern Italy, northwestern Germany, and eastern Austria (Figure 3D).

### 3.5. Ecological Niches Suitable for Main Vectors and Hosts

By comparing the kappa values and F1 scores of the BRT models with different sampling ratios of main vectors, the model that applied 1:1 sampling ratio for each vector was chosen as the optimal model (Appendix A). The AUC of these selected models ranged from 0.895 to 0.993, indicating decent performance (Appendix A). Ecological factors contributed significantly, with the variable of land cover type having the greatest effect on the distribution of vectors, while climate covariates significantly contributed to the distribution of hosts (Appendix A). The predicted distribution map accurately captured the actual distribution areas for each primary vector and host (Appendix A), and additionally identified previously uninvestigated areas (Figure 5 and Appendix A).

### 3.6. Ecological Niches Suitable for USUV

The modeling of USUV infection demonstrated AUCs ranging from 0.893 to 0.993. Random grid sampling exhibited superior performance compared to background cell sampling, and BRT and RF models outperformed the LASSO model (Appendix A). Optimal performance was obtained when a threshold of 900 km^2^ and a sampling ratio of 1:5 were employed in the BRT, RF, and LASSO models. According to the optimal model, the mosquito index had the most significant impact on the distribution of USUV infections, especially in the BRT model. RCs were demonstrated to exceed 50%, indicating vectors’ critical role in affecting USUV distribution (Appendix A). The final model was obtained by constructing a composite learner through the bagging method, based on which the areas at potential risk for USUV infections were mapped, which covered the known distribution of USUV (Figure 6A) and identified additional regions at potential risk that were previously uninvestigated, mainly in Africa and the Middle East (Figure 6B). The projected regions at potential risk of USUV infections (low-to medium-risk areas and high-risk areas) covered a potential suitable habitat of 1.80 million km^2^ inhabited by nearly 1.04 billion people. It is noteworthy that Europe had significantly higher proportions of high-risk areas and populations compared to Africa and the Middle East (Table 2).

## 4. Discussion

Reports of human infection with USUV by an increasing number of countries represent a potential public health threat that is recognized. The timeline of USUV-infected cases revealed that since the initial occurrence of a confirmed USUV case in 1981 in the Central African Republic, a second emergency arose following an interval of 23 years in Burkina Faso in 2004 [10]. Subsequently, five years after these incidents, Europe reported its first confirmed case of USUV [6], with over 200 human infections documented across Europe to the end of 2023 [22,25,26,37,38,39,40]. Here, we present a comprehensive investigation of USUV infections based on systematic assembling of publicly available data, demonstrating the spatiotemporal characteristics of USUV infections in humans, vectors, and animals.

Our study has systematically mapped the spatial distribution of USUV infections in humans (a total of 235 cases), as well as across 20 vector species and 175 animal host species. However, it is important to note that the reported human cases may only represent a fraction of actual infections due to several factors: a significant number of individuals infected with USUV remain asymptomatic, and many cases are incidentally detected during routine WNV screening among asymptomatic blood donors [41,42]. Several studies have even reported higher prevalence rates for asymptomatic USUV infection compared to WNV [29,39,43]. Therefore, it is justified to propose an intensified surveillance of USUV infections in Europe. Furthermore, our study revealed that mosquito index was the most significant factor contributing to the occurrence of USUV infection, highlighting the crucial role of mosquito vectors in the spread of this virus. Previous studies solely considered mosquitoes as vectors for USUV; however, in 2023, ticks were first identified to be infected by USUV (GenBank no. OP921077, OP921079-OP921082), suggesting the potential involvement of vectors other than mosquitoes in the transmission of USUV and warranting further investigation.

Currently, modeling studies have investigated contributors to the distribution of USUV. For example, Rubel et al. revealed that both bird immune status and environmental temperature play significant roles in in determining the transmission dynamics of USUV [31]. Cheng et al. indicated that the minimum temperature of the coldest month had the strongest contribution to the distribution of USUV [30]. Our current model had the advantage of incorporating a full consideration of variables at multiple steps and therefore to provide improved estimates of high-risk areas for the transmission of USUV and people at risk, which is essential for prioritizing health services in the future. The mosquito index was applied as a highly efficient vector factor, and was revealed to play a pivotal role in determining the distribution of USUV. Moreover, the percentage coverage of urban construction land emerges as a primary determinant in vector distribution, exhibiting a close association with urbanization processes akin to biological invasions [44]. Urbanization has been demonstrated to play a crucial role in the transmission of certain mosquito-borne diseases, primarily those transmitted by Aedes mosquitoes such as dengue, Zika, and chikungunya. This is due to its contribution in creating favorable breeding sites through human-made containers, increasing the likelihood of interactions between vectors and humans due to higher population densities, facilitating spatial spread through the movement of people and goods, as well as enabling expansion for certain wild species [45]. However, the impact of urbanization on mosquito-borne diseases can vary significantly depending on the specific mosquito species and diseases in question, for instance, urbanization has the potential to diminish the habitat suitability for malaria vectors (e.g., *Anopheles* mosquitoes), which typically prefer rural or semi-rural environments with natural water sources. The process of urban development often results in the removal of such habitats, thereby reducing the distribution of malaria vectors in numerous cases [46].

Our study has several limitations. First, in the modeling analysis, despite achieving maximum matching between the occurrence of USUV and the time range of other variables, inconsistency remains, potentially introducing bias into the modeling results. Secondly, we utilized the average value of covariates over multiple years within polygons to present the modeling value; however, this approach may lead to ecological fallacy when covariates are distributed unevenly within a polygon and also overlooks possible temporal evolution. Therefore, while the distribution can be considered indicative of general epidemic trends of USUV, it cannot be interpreted as an accurate reflection of prevalence. Thirdly, climate, land use type, socioeconomic covariates were used twice to model the distribution of vectors and hosts, as well as the distribution of USUV, and this strategy could have led to collinearity or overfitting of the model. However, it was also reasonable to some extent, because the potential mechanisms of the same variable might vary in different models. For example, in the predicted distribution model of vectors and hosts, urban construction land may affect their living environment and habitat, thereby affecting their breeding, reproduction, and density, while in the predicted distribution model of USUV, it could affect human activities and increase human exposure risks. To exclude the influence of high collinearity among variables, we also conducted a collinearity diagnosis of variables by calculating their correlation coefficients (threshold of 0.75) in each stage. Fourthly, we only predicted the occurrence of three mosquitoes with the highest positivity rates among the 20 mosquitoes and the three hosts with the highest “weighted importance” among the 166 bird hosts generating the mosquito index and bird index. We aimed to incorporate vector and host factors into the model by modeling the most representative species, which could simplify the model to some extent; however, this strategy did not consider all vectors and hosts. Although secondary vectors and hosts play a limited role in USUV transmission, our study design might have had an impact on the results, especially in areas dominated by other vectors or hosts, where the transmission risk of USUV might be underestimated.

## 5. Conclusions

Our research indicates that the impact of USUV is mainly concentrated in Europe, and the natural cycle of USUV involves a complex and diverse range of vector and host species. The distribution of USUV is influenced by multiple factors, with vectors playing the most important role. Moreover, the predictive risk map generated through modeling could effectively guide future surveillance efforts for USUV infections, especially for countries located within high-risk areas and those that have not yet conducted surveillance activities.

## Figures and Tables

**Figure 1 viruses-16-01606-f001:**
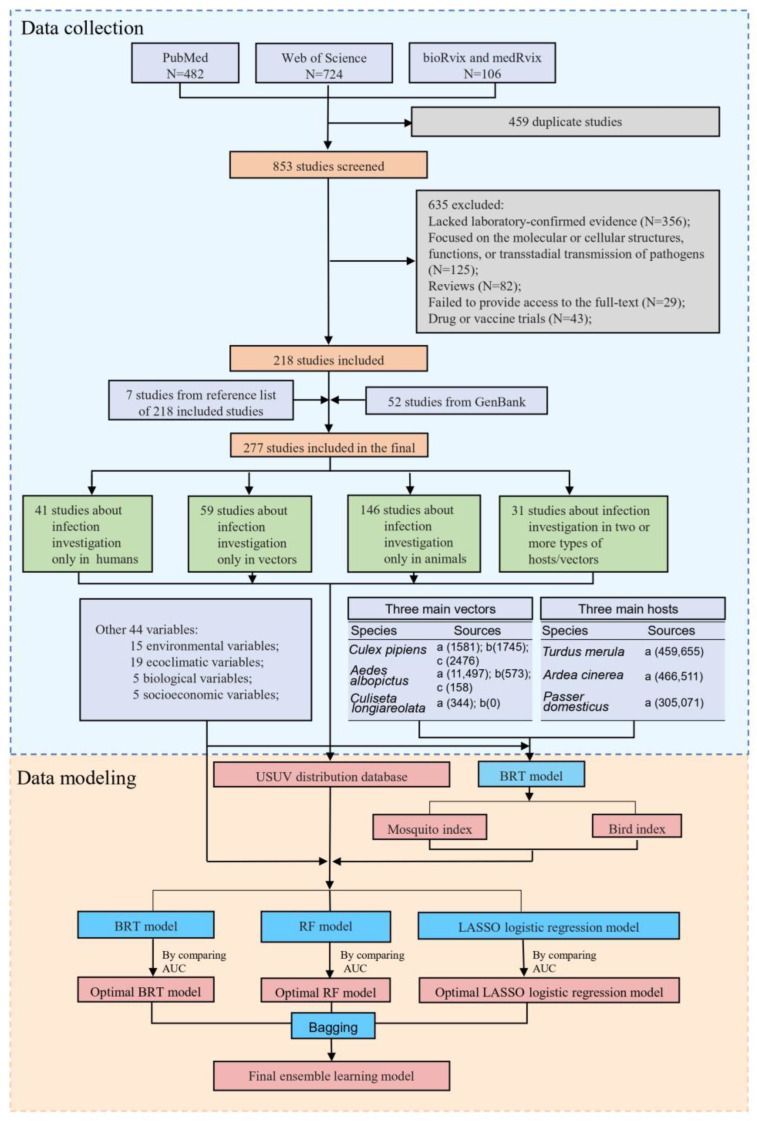
Flow diagram of the literature review and data modeling. The entire process can be divided into two parts, data collection and data modeling. IN the data sources of vectors and hosts, a presents GBIF, b presents VectorMap, and c represents other articles. The numbers in numbers parentheses for the sources of data of three main vectors and three main hosts represent the corresponding number of records. BRT: boosted regression tree. RF: random forest. LASSO: least absolute shrinkage and selection operator.

**Figure 2 viruses-16-01606-f002:**
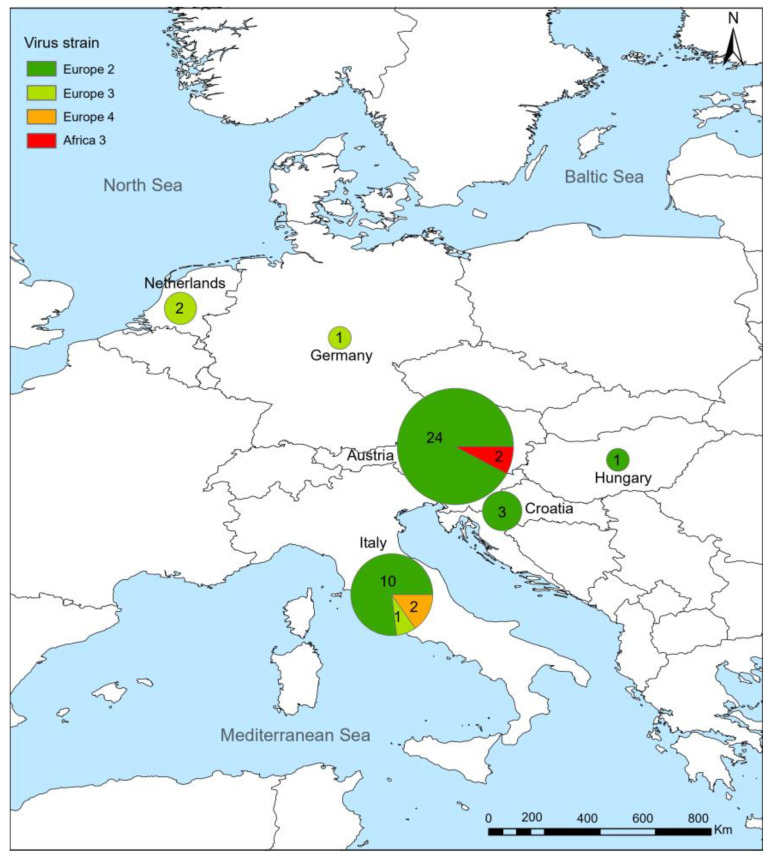
The geographical distribution of USUV strains reported in humans. Locations and numbers of USUV strains are shown, and all strains were confirmed by sequencing.

**Figure 3 viruses-16-01606-f003:**
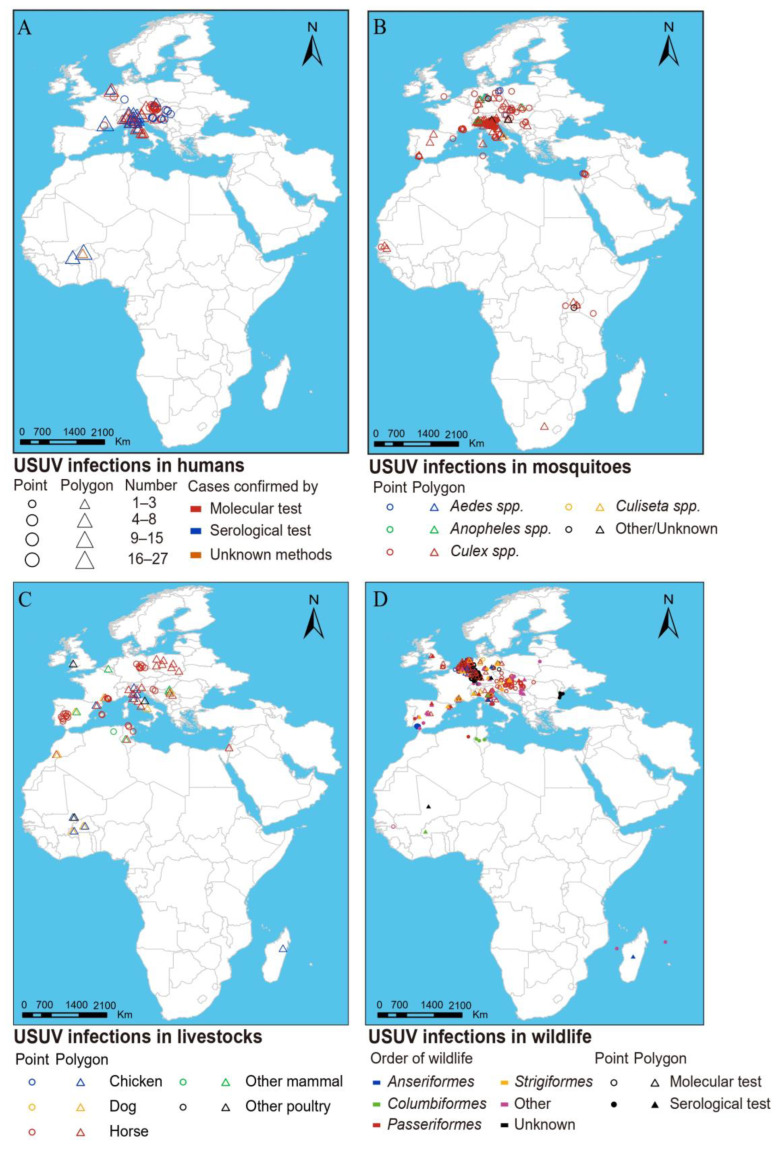
European, African, and Middle Eastern distribution of USUV detection in vectors and hosts. (**A**) Humans. (**B**) Vectors. (**C**) Livestock. (**D**) Wildlife. Positive detections in humans and wildlife were determined by molecular and serological tests, positive detections in vectors were determined only by molecular tests, and positive detections in livestock were determined only by serological tests.

**Figure 4 viruses-16-01606-f004:**
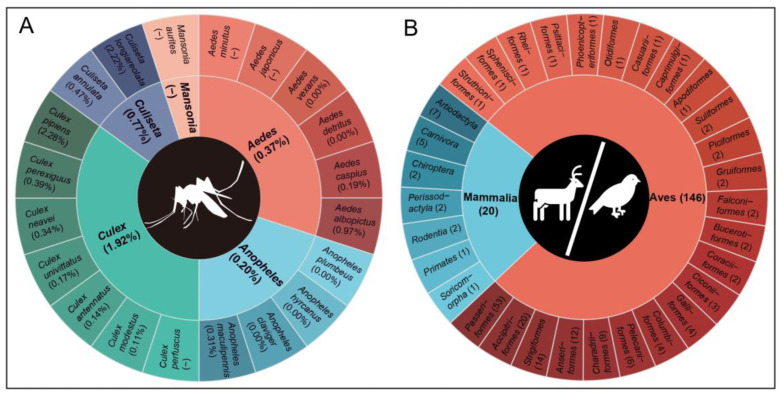
The number of vectors and animal species from which USUV was detected. (**A**) USUV detected in vectors, including 20 mosquito species, further classified by genus. Numbers in parentheses represent the positive rate of USUV infection. (**B**) USUV detected in animals, including 146 avian species and 20 mammalian species, further classified by order. Numbers in parentheses represent the exact number of species.

**Figure 5 viruses-16-01606-f005:**
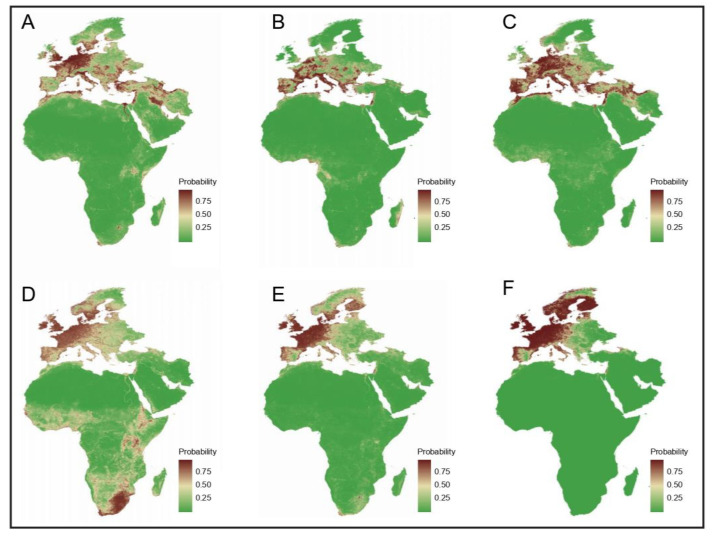
Predicted geographical distribution of the three main vectors and three main animal hosts. (**A**) *Culex pipiens*. (**B**) *Aedes albopictus*. (**C**) *Culiseta longiareolata*. (**D**) *Ardea cinerea*. (**E**) *Passer domesticus*. (**F**) *Turdus merula*. Occurrence was predicted by the BRT model, for which randomly selecting “control” from 30 to 3000 km from the occurrence cells with 1:1 sampling ratio was applied. The color legend represents a scale of the relative probability that the species occurred in that location from 0 (green, low) to 1 (red, high).

**Figure 6 viruses-16-01606-f006:**
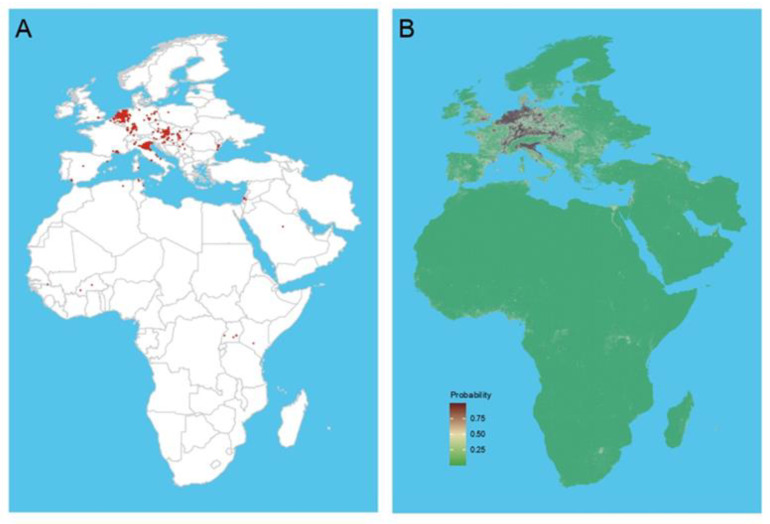
The actual recorded and predicted distributions of USUV. (**A**) The actual recorded occurrence of USUV. The red dot indicates the actual USUV occurrence of point data and polygon data with an area below 900 km^2^. (**B**) The predicted occurrence of USUV. The occurrence was predicted by composite learner, for which the bootstrap aggregation (bagging) method was applied and the best BRT, RF, and LASSO models weighted based on their AUCs. The color legend represents a scale of the relative probability that the USUV occurred in that location from 0 (green, low) to 1 (red, high).

**Table 1 viruses-16-01606-t001:** Demographic characteristics and clinical manifestations of USUV infection in humans.

	All (235)	Italy (141)	Austria (27)	Burkina Faso (20)	France (16)	Croatia (11)	Other Countries (20)	*p*
Period (*n*, %)								<0.001 ^b^
Before 2011	37 (16)	35 (26)	0 (0)	1 (5)	0 (0)	0 (0)	1 (5)	
After 2012	194 (84)	102 (74)	27 (100)	19 (95)	16 (100)	11 (100)	19 (95)	
Unknown ^a^	4	4	0	0	0	0	0	
Sex (*n*, %)								0.218 ^b^
Male	67 (67)	23 (68)	21 (81)	11 (58)	1 (−)	2 (−)	9 (53)	
Female	33 (33)	11 (32)	5 (19)	8 (42)	0 (−)	1 (−)	8 (47)	
Unknown ^a^	135	107	1	1	15	8	3	
Age group (*n*, %)								<0.001 ^b^
<40	32 (56)	4 (31)	0 (−)	18 (95)	1 (−)	2 (−)	7 (41)	
41–70	22 (39)	8 (62)	0 (−)	1 (5)	0 (−)	3 (−)	10 (59)	
>70	3 (5)	1 (8)	1 (−)	0 (0)	0 (−)	1 (−)	0 (0)	
Unknown/Unclear ^a^	178	128	26	1	15	5	3	
Month (*n*, %)								<0.001 ^b^
Before June	1 (1)	0 (−)	0 (0)	0 (0)	0 (−)	0 (−)	1 (−)	
June–September	70 (85)	16 (62)	26 (100)	19 (100)	0 (−)	6 (−)	3 (−)	
After September	11 (13)	10 (38)	0 (0)	0 (0)	1 (−)	0 (−)	0 (−)	
Unknown ^a^	153	115	1	1	15	5	16	
Under immunosuppressive status (*n*, %)								<0.001 ^b^
Yes	4 (2)	3 (3)	0 (0)	0 (0)	0 (−)	1 (−)	0 (0)	
No	177 (98)	105 (97)	27 (100)	20 (100)	1 (−)	6 (−)	18 (100)	
Unknown ^a^	54	33	0	0	15	4	2	
Virus strain (*n*, %)								0.003 ^b^
Europe 2	38 (83)	10 (77)	24 (92)	0 (−)	0 (−)	3 (−)	1 (−)	
Europe 3	4 (9)	1 (8)	0 (0)	0 (−)	0 (−)	0 (−)	3 (−)	
Europe 4	2 (4)	2 (15)	0 (0)	0 (−)	0 (−)	0 (−)	0 (−)	
Africa 3	2 (4)	0 (0)	2 (8)	0 (−)	0 (−)	0 (−)	0 (−)	
Untyped ^a^	189	128	1	20	16	8	16	
Co-infection (*n*, %)								<0.001 ^b^
Yes	48 (39)	35 (53)	1 (5)	0 (−)	1 (7)	4 (−)	7 (47)	
WNV infection	45 (94)	35 (100)	1 (−)	0 (−)	1 (−)	1 (−)	7 (−)	
TOSV infection	3 (6)	0 (0)	0 (−)	0 (−)	0 (−)	3 (−)	0 (−)	
No	74 (61)	31 (47)	18 (95)	0 (−)	14 (93)	3 (−)	8 (53)	
Undetected ^a^	113	75	8	20	1	4	5	
Symptoms (*n*, %)								
Asymptomatic	94−96 (70−71)	53 (68)	24−25 (92−96)	0 (−)	0 (−)	1−2 (10−20)	16 (84)	
Encephalitis/Meningitis	27−29 (20−21)	15 (19)	1−2 (4−8)	0 (−)	0 (−)	9−10 (90−100)	2 (11)	
Fever	17−31 (13−23)	10−15 (13−19)	1−2 (4−8)	1 (−)	0 (−)	3−10 (30−100)	2−3 (11−16)	
Headache	14−28 (10−21)	10−15 (13−19)	0−1 (0−4)	0 (−)	0 (−)	3−10 (30−100)	1−2 (5−11)	
Rash	5−19 (4v14)	3−8 (4−10)	1−2 (4−8)	0 (−)	0 (−)	0−7 (0−70)	1−2 (5−11)	
Asthenia	6−13 (4−8)	6−11 (8−14)	0−1 (0−4)	0 (−)	0 (−)	0−7 (0−70)	0−1 (0−5)	
Myalgia	5−12 (4−9)	5−10 (6−13)	0−1 (0−4)	0 (−)	0 (−)	0−7 (0−70)	0−1 (0−5)	
Unknown ^a^	100	63	1	19	15	1	1	

WNV: West Nile virus. TOSV: Toscana virus. ^a^: Cases with incomplete information were excluded from the percentage calculation. ^b^: *p*-values of intergroup differences were calculated using Fisher probabilities. All numbers representing percentages were rounded to the nearest integer. Data with a total fewer than 10 or a denominator less than 10 were not included in percentage calculations. Other countries included the Netherlands, Hungary, Germany, Czech Republic, Republic of South Africa, and Central African Republic. The detailed results of each country are shown in Appendix A. When calculating the frequency of each clinical symptom, we divided publications into two groups, case reports and case series. A case report describes clinical features of a single patient in detail, for which it is reasonable to assume unmentioned symptoms are absent. In contrast, a case series summarizes clinical characteristics of a group of confirmed patients, for which it is unclear if unmentioned symptoms are truly absent from the whole group or just rare, especially when the group is large. For case series, we therefore made a conservative assumption that the frequency of an unreported symptom could vary from 0 to the minimum frequency of all reported symptoms. Consequently, we report a range for each symptom if relevant data involve case series. For the recording of clinical symptoms/signs, only patients with confirmed infections were used.

**Table 2 viruses-16-01606-t002:** The area and population size of USUV occurrence risk predicted by the final model.

		All	Europe	Africa	Middle East	*p*
Population size (million)	High-risk area	246.3	229.4 (93.14%)	4.8 (1.95%)	12.1 (4.91%)	<0.001
	Low- to medium-risk area	795.2	216.5 (27.23%)	460.7 (57.94%)	118.0 (14.84%)	
Area (1000 km^2^)	High-risk area	533.9	529.9 (99.25%)	0.7 (0.13%)	3.4 (0.64%)	<0.001
	Low- to medium-risk area	1261.8	1012.9 (80.27%)	177.9 (14.10%)	70.9 (5.62%)	

## Data Availability

The datasets used and/or analyzed during the current study are available from the corresponding author on reasonable request.

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
