# Peer review of "Epidemiology and Ecology of Usutu Virus Infection and Its Global Risk Distribution"

_viruses, 2024, doi:10.3390/v16101606_

Round 1
Reviewer 1 Report
Comments and Suggestions for Authors
The manuscript presents an important study on the epidemiology and ecological risk of USUV across regions, with a focus on Africa and Europe. The work is timely and relevant, given the growing public health concern posed by some neglected mosquito-borne flaviviruses. The study covers a wide range of data sources, incorporating ecological, environmental, and socio-economic factors into the risk models. The use of ensemble learning (BRT, RF, LASSO) for predicting USUV infection risk is a key strength. Overall, the manuscript is scientifically sound and offers valuable insights for guiding public health strategies, particularly in regions that lack current surveillance efforts, but a few revisions are necessary to improve clarity and enhance the overall impact of the study.
1. The introduction is informative, but it could benefit from a more explicit statement of research questions or hypotheses, to guide the reader on the novelty and objectives of the study. For example, what are the gaps in current knowledge of USUV epidemiology, and why is this modelling effort necessary?
2. The manuscript provides a detailed mapping of the distributions of mosquito vectors such as Aedes albopictus and Culex pipiens—species that have been studied in previous research (e.g. Kraemer et al. Past and future spread of the arbovirus vectors Aedes aegypti and Aedes albopictus. Nat Microbiol 2019. https://doi.org/10.1038/s41564-019-0376-y). The authors should clarify the rationale for redoing the vector mapping despite the availability of existing datasets. Emphasizing the benefit of using an updated, virus-specific, integrated modelling approach will reinforce the added value of this study.
3. The approach of using environmental, ecoclimatic, biological, and socioeconomic covariables twice—first to model the distribution of vectors and hosts, and then to model the ecological niche of USUV—could raise concerns about collinearity or overfitting due to the double use of these variables. However, this approach can be justified. The authors may acknowledge the potential concerns regarding the double use of covariables and briefly explain how their modelling approach mitigates these risks. They could also highlight the hierarchical structure of the modelling framework or discuss how regularization methods were employed to address any issues related to collinearity or overfitting. This would strengthen the methodological rigour and transparency of the study.
4. The manuscript limits the mosquito and bird species in the risk index to three primary vectors and hosts. While this simplifies the modelling, the rationale for excluding other potentially important species should be briefly discussed. This will help contextualize the limitations of the model.
5. The discussion about urbanization highlights the role of urbanization in promoting mosquito-borne disease transmission (e.g. dengue, Zika, and chikungunya) by creating artificial breeding sites (e.g., water containers). However, this statement may need refinement to clarify that the effects of urbanization can vary significantly depending on the specific mosquito species and disease in question. For example, urbanization can reduce the habitat suitability for malaria vectors (e.g., Anopheles mosquitoes), which typically prefer rural or semi-rural environments with natural water sources. Urban development often leads to the removal of such habitats, thus decreasing the distribution of malaria vectors in many cases.
6. Are there any relevant Chinese publications/reports of imported or local USUV infections in China?
Comments on the Quality of English LanguageOverall, the manuscript is well-written, but some sections mentioned above could be improved to make the content more accessible and impactful.
Author Response
Reviewer 1:
- The introduction is informative, but it could benefit from a more explicit statement of research questions or hypotheses, to guide the reader on the novelty and objectives of the study. For example, what are the gaps in current knowledge of USUV epidemiology, and why is this modelling effort necessary?
[Response] We greatly appreciate your valuable suggestion. We have supplemented the research questions and the necessity of our modeling effort in the revised Introduction section as: “With the increasing global attention on USUV infection and advancements of molecular diagnostic technology, there has been a surge in research focusing on local surveillance of USUV infection as well as modeling analysis[30, 31]. However, current research efforts are limited to specific countries or regions, leaving the worldwide distribution of USUV infections in vectors and hosts and associated risk burden uncertain.
To comprehensively analyze the diversity, distribution, and potential risk associated with USUV infection, we conducted a systematic literature review on the global occurrence of USUV infections in vectors, animals, and humans. Based on which, we developed ma-chine learning models to assess environmental suitability of USUV and estimate the glob-al risk distribution associated with USUV infections.” (Page 2, lines 63-72)
- The manuscript provides a detailed mapping of the distributions of mosquito vectors such as Aedes albopictus and Culex pipiens—species that have been studied in previous research (e.g. Kraemer et al. Past and future spread of the arbovirus vectors Aedes aegypti and Aedes albopictus. Nat Microbiol 2019. https://doi.org/10.1038/s41564-019-0376-y). The authors should clarify the rationale for redoing the vector mapping despite the availability of existing datasets. Emphasizing the benefit of using an updated, virus-specific, integrated modelling approach will reinforce the added value of this study.
[Response] Many thanks for your valuable suggestion. According to your suggestion, we add description of clarifying the rationale for redoing the vector mapping despite the availability of existing datasets as: “The potential distributions of Aedes albopictus and Culex pipiens have been previously investigated[33, 34]. However, considering the updated data since 2014 and 2022, and the different regions involved in our study compared to previous studies, as well the incorporation of additional variables in our models, we have supplemented the occurrence data in the original dataset and conducted a new model.” (Page 3, lines 141‒145)
- The approach of using environmental, ecoclimatic, biological, and socioeconomic covariables twice—first to model the distribution of vectors and hosts, and then to model the ecological niche of USUV—could raise concerns about collinearity or overfitting due to the double use of these variables. However, this approach can be justified. The authors may acknowledge the potential concerns regarding the double use of covariables and briefly explain how their modelling approach mitigates these risks. They could also highlight the hierarchical structure of the modelling framework or discuss how regularization methods were employed to address any issues related to collinearity or overfitting. This would strengthen the methodological rigor and transparency of the study.
[Response] We greatly appreciate your valuable suggestion. As you said, it is indeed possible for occurring collinearity or overfitting of the model. We have added a discussion on the issue of the same variable participating in different modeling processes in the revised limitation section as: “Thirdly, climate, land use type, socio-economic covariables were used twice to model the distribution of vectors and hosts, as well as the distribution of USUV, this strategy could lead to collinearity or overfitting of the model. However, it was also reasonable to some extent, because the potential mechanisms of the same variable might vary in different models. For example, in the predicted distribution model of vectors and hosts, urban construction land may affect their living environment and habitat, thereby affecting their breeding, reproduction, and density, while in the predicted distribution model of USUV, it could affect human activities and increase human exposure risks. To exclude the influence of high collinearity among variables, we also conducted collinearity diagnosis of variables by calculating their correlation coefficients (threshold was 0.75) in each stage.” (Page 14, lines 383-393)
- The manuscript limits the mosquito and bird species in the risk index to three primary vectors and hosts. While this simplifies the modelling, the rationale for excluding other potentially important species should be briefly discussed. This will help contextualize the limitations of the model.
[Response] We appreciate your valuable suggestions. According to your suggestion, we add description of choosing main vectors and main hosts, and we further add discussion about excluding other potentially important species in limitations.
The details of the modified content in the manuscript are as follows: “The main vector species and host species of USUV currently lack a systematic consensus in various studies. Therefore, we established the USUV spatial distribution database by collecting data from all relevant literature to determine the primary vector and host species. In this database, only molecular detection was available for the positive infection rate in vectors, while molecular detection and serological detection could be used in hosts, and the total sample size and positive infection rate of molecular and serological detections in host were different, so we determined the primary and secondary vectors by the positive infection rate, and determine the primary and secondary hosts by calculating " weighted importance ". In specific, based on positive detection rate exceeding 0.50%, we identified three mosquito species (Culex pipiens, Aedes albopictus, and Culiseta longiareolata) as the primary vectors (Appendix A Table 5). And by calculating “weighted importance” (Appendix A page 3), we determined three bird hosts (Turdus merula, Passer domesticus, and Ardea cinerea) (Appendix A Table 6) as the primary hosts.” (Page 3, lines 123-136)
“Fourthly, we only predicted the occurrence of three mosquitoes with the highest positivity rates among the 20 mosquitoes and the three hosts with the highest “weighted importance” among the 166 bird hosts to generate mosquito index and bird index. We aim to incorporated vector and host factors into the model by modeling the most representative species, which could simplify the model to some extent, however, this strategy did not consider all vectors and hosts, although secondary vectors and hosts play a limited role in USUV transmission, our study might have an impact on the results, especially in areas dominated by other vectors or hosts, the transmission risk of USUV might be underestimated.” (Page 14, lines 393-401)
- The discussion about urbanization highlights the role of urbanization in promoting mosquito-borne disease transmission (e.g. dengue, Zika, and chikungunya) by creating artificial breeding sites (e.g., water containers). However, this statement may need refinement to clarify that the effects of urbanization can vary significantly depending on the specific mosquito species and disease in question. For example, urbanization can reduce the habitat suitability for malaria vectors (e.g., Anopheles mosquitoes), which typically prefer rural or semi-rural environments with natural water sources. Urban development often leads to the removal of such habitats, thus decreasing the distribution of malaria vectors in many cases.
[Response] Many thanks for your helpful and professional suggestion. We have supplemented this point in the Discussion section as: “Urbanization has been demonstrated to play a crucial role in the transmission of certain mosquito-borne diseases, primarily those transmitted by Aedes mosquitoes such as dengue, Zika, and chikungunya. This is due to its contribution in creating favorable breeding sites through human-made containers, increasing the likelihood of interactions between vectors and humans due to higher population densities, facilitating spatial spread through the movement of people and goods, as well as enabling expansion for certain wild species[45]. However, the impact of urbanization on mosquito-borne diseases can vary significantly depending on the specific mosquito species and diseases in question, for instance, urbanization has the potential to diminish the habitat suitability for malaria vectors (e.g., Anopheles mosquitoes), which typically prefer rural or semi-rural environments with natural water sources. The process of urban development often results in the removal of such habitats, thereby reducing the distribution of malaria vectors in numerous cases[46]” (Page 13, lines 362-374)
- Are there any relevant Chinese publications/reports of imported or local USUV infections in China?
[Response] Many thanks for your comment. As of now, there have been no reported cases of imported or local USUV infections in China.

Reviewer 2 Report
Comments and Suggestions for Authors
The manuscript "Epidemiology and ecology of Usutu virus infection and its global risk distribution" is an article in two parts. It first conducts a literature review on Usutu virus to consider human infections, arthropod vectors and infection in animals. It then uses ecological niche data on selected vectors and hosts to determine risk of distribution in Europe and Africa. Usutu virus has been thoroughly reviewed in recent years (see Simonin 2024 Viruses 16, 599; Angeloni et al 2023 Euro Surveill 28, 2200928; Vilibic-Cavlek et al 2020 Pathogen 9, 699) so the inclusion of risk modelling, whilst unorthodox, is probably necessary. However, the article requires improvement so that the aims are clear and the conclusions reflect the analysis presented. The following major concerns should be addressed and a list of minor corrections/clarifications are listed below.
1. The authors state throughout that they are assessing "global risk distribution". Firstly, this needs clarifying. Do they mean the maximum extent of Usutu distribution or risk of emergence? Secondly, when using terms such as global and worldwide, this presumes that they will consider the whole world and not just Europe and Africa as they seem to do. The manuscript should be revised to reflect this. For example, why was North America not considered as it contains both arthropod vectors and vertebrate reservoirs.
2. The authors should give a full justification for their selection of Cx pipiens, Ae albopictus and Cs longiareolata as the key vector species. No argument with Cx pipiens but the evidence for the other two is tenuous at best. Likewise, the selection of the three vertebrate host requires justification in the text rather than repeated reference to supplementary files (of which there are over 100 pages).
3. The quality of the figures, particularly Figures 3, 4 and 5 should be improved so that text is legible and the locations are clearly discernible. The legend of Figure 2 needs to state clearly that this refers to USUV strains detected in humans and this should also be stated in the text (lines209-210). The maps in Figure 3 should be consistent and show the same geographical area, ideally all of Europe and North Africa.
4. The Discussion should be improved to emphasise the importance of the findings. To state in line 324 "Out results suggest that USUV has established stable transmission cycle in Europe, acting as a high-risk region for USUV infection, despite of its originating from Africa." is not original and as the study was based on a literature review, not strictly the results of this study. The following paragraph (lines 334-350) does not state anything new and could be removed. Finally, the Conclusions do not state anything conclusive. What exactly does the literature review find that was not already known and what did the risk distribution model find other than USUV can be found in the regions where it is found?
Minor comments
Line 16, why is USUV a "significant threat to public health". Compared to other arthropod borne viruses, it is not.
Line 33. USUV is now classified in the genus Orthoflavivirus by ICTV.
Line 34. The reference used to justify that USUV is an emerging virus is from 1998, suggest using something a little more contemporary.
Line 46. Turdus merula, and all other latin names should be in italics.
Line 66. what is "USUA"
Table 1. the formatting of the second row requires alignment.
Figure 3, suggest revising to "European, African and Middle Eastern distribution of USUV..."
Line 238, revise to "In total, USUV was detected in 20 mosquito species, including ..."
Figure 5, revise to "(A) Culex pipiens. (B) Aedes albopictus. (C) Culiseta longiareolata. (D) Ardea cinerea... etc. Inclusion of "vector of" and "host of" is redundant.
Line 302. What are "risky" regions? Suggest revising to something more accurate.
Comments on the Quality of English LanguageSome improvements, as listed to the authors, have been suggested.
Author Response
- The authors state throughout that they are assessing "global risk distribution". Firstly, this needs clarifying. Do they mean the maximum extent of Usutu distribution or risk of emergence? Secondly, when using terms such as global and worldwide, this presumes that they will consider the whole world and not just Europe and Africa as they seem to do. The manuscript should be revised to reflect this. For example, why was North America not considered as it contains both arthropod vectors and vertebrate reservoirs.
[Response] We greatly appreciate your valuable comments. Following your suggestion, we have supplemented the necessary explanations of the modeling areas in the Method section as: “Although we aimed to explore the global risk distribution of USUV infections, our literature search and data extraction on a global scale revealed that USUV infection has only been reported in Europe, Africa, and some regions of the Middle East. Therefore, we hypothesize that USUV has not yet established a complete transmission chain in other regions where both its arthropod vectors and vertebrate reservoirs exist, such as North America. To present the current areas with potential risk, our modeling area was limited to Europe, Africa, and the Middle East” (Page 3, Lines 115‒121)
- The authors should give a full justification for their selection of Cx pipiens, Ae albopictus and Cs longiareolata as the key vector species. No argument with Cx pipiens but the evidence for the other two is tenuous at best. Likewise, the selection of the three vertebrate host requires justification in the text rather than repeated reference to supplementary files (of which there are over 100 pages).
[Response] Many thanks for your valuable suggestion. Following your suggestion, we have supplemented the description regarding the selection of main vectors and hosts in the Methods section of the revised main text as: “The main vector species and host species of USUV currently lack a systematic consensus in various studies. Therefore, we established the USUV spatial distribution database by collecting data from all relevant literature to determine the primary vector and host species. In this database, only molecular detection was available for the positive infection rate in vectors, while molecular detection and serological detection could be used in hosts, and the total sample size and positive infection rate of molecular and serological detections in host were different, so we determined the primary and secondary vectors by the positive infection rate, and determine the primary and secondary hosts by calculating " weighted importance ". In specific, based on positive detection rate exceeding 0.50%, we identified three mosquito species (Culex pipiens, Aedes albopictus, and Culiseta longiareolata) as the primary vectors (Appendix A Table 5). And by calculating “weighted importance” (Appendix A page 3), we determined three bird hosts (Turdus merula, Passer domesticus, and Ardea cinerea) (Appendix A Table 6) as the primary hosts.” (Page 3, lines 123‒136)
In addition, we have further discussed this issue about excluding other potentially important species in the limitations section as: “Fourthly, we only predicted the occurrence of three mosquitoes with the highest positivity rates among the 20 mosquitoes and the three hosts with the highest “weighted importance” among the 166 bird hosts to generate mosquito index and bird index. We aim to incorporated vector and host factors into the model by modeling the most representative species, which could simplify the model to some extent, however, this strategy did not consider all vectors and hosts, although secondary vectors and hosts play a limited role in USUV transmission, our study might have an impact on the results, especially in areas dominated by other vectors or hosts, the transmission risk of USUV might be underestimated” (Page 14, lines 393‒401)
- The quality of the figures, particularly Figures 3, 4 and 5 should be improved so that text is legible and the locations are clearly discernible. The legend of Figure 2 needs to state clearly that this refers to USUV strains detected in humans and this should also be stated in the text (lines 209-210) . The maps in Figure 3 should be consistent and show the same geographical area, ideally all of Europe and North Africa
[Response] We appreciate your helpful suggestions. We have improved the quality of the figures to make text is legible and the locations are clearly discernible in these figures, and have also made modifications to the legend of Figure 2 and related text as your suggestions. In addition, we have updated the Figure 3 to maintain the same background. (Figure 2, Figure 3, Figure 4, and Figure 5)
- The Discussion should be improved to emphasize the importance of the findings. To state in line 324 "Out results suggest that USUV has established stable transmission cycle in Europe, acting as a high-risk region for USUV infection, despite of its originating from Africa." is not original and as the study was based on a literature review, not strictly the results of this study. The following paragraph (lines 334-350) does not state anything new and could be removed. Finally, the Conclusions do not state anything conclusive. What exactly does the literature review find that was not already known and what did the risk distribution model find other than USUV can be found in the regions where it is found?
[Response] Thanks for your valuable suggestion. Following your suggestion, we have removed the paragraph (lines 334-350), and rewritten the relevant parts of the discussion as: “Our study has systematically mapped the spatial distribution of the USUV infections in humans (a total of 235 cases), across 20 vector species and 175 animal host species. However, it is important to note that the reported human cases may only represent a fraction of actual infections due to several factors: a significant number of individuals infected with USUV remain asymptomatic, and many cases are incidentally detected during routine WNV screening among asymptomatic blood donors[41, 42]. Several studies have even reported higher prevalence rates for asymptomatic USUV infection compared to WNV[29, 39, 43]. Therefore, it is justified to propose an intensified surveillance of USUV infections in Europe. Furthermore, our study revealed that mosquito index was the most significant factor contributing to the occurrence of USUV infection, highlighting the crucial role of mosquito vectors in the spread of this virus. Previous studies solely considered mosquitoes as vectors for USUV; however, in 2023, ticks were first identified to be infected by USUV (GenBank No. OP921077, OP921079-OP921082), suggesting the potential involvement of other vectors other than mosquito in the transmission of USUV and warranting further investigation” (Page 13, lines 335-349)
Also, we have updated the conclusion following your suggestion as: “Our research indicates that the impact of USUV is mainly concentrated in Europe, and the natural cycle of USUV involves a complex and diverse range of vector and host species. The distribution of USUV is influenced by multiple factors, with vector playing the most important role. Moreover, the predictive risk map generated through modeling could effectively guide future surveillance efforts for USUV infections, especially for countries located within high-risk areas and those that have not yet conducted surveillance activities.” (Page 14, lines 403-409)
- Line 16, why is USUV a "significant threat to public health". Compared to other arthropod borne viruses, it is not.
[Response] Many thanks for your correction. We have changed to “posing a potential threat to public health”. (Page 1, line 16)
- Line 33. USUV is now classified in the genus Orthoflavivirus by ICTV.
[Response] Thanks for your suggestion. According to your suggestion, we make modifications to this statement. (Page 1, line 33 & Page 2, line 92)
- Line 34. The reference used to justify that USUV is an emerging virus is from 1998, suggest using something a little more contemporary
[Response] Many thanks for your suggestion. We have added a new reference in this line. (Page 1, line 34)
- Line 46. Turdus merula, and all other latin names should be in italics.
[Response] Many thanks for your suggestion. We have modified it as suggested. (Page 2, line 46; Page 3, line 146)
- Line 66. what is "USUA"
[Response] Many thanks for your suggestion. We have modified it to USUV. (Page 2, line 70)
- Table 1. the formatting of the second row requires alignment.
[Response] Many thanks for your suggestion. We have modified it as suggested. (Page 7, Table 1)
- Figure 3, suggest revising to "European, African and Middle Eastern distribution of USUV..."
[Response] Many thanks for your suggestion. We have modified it as suggested. (Page 9, line 236)
- Line 238, revise to "In total, USUV was detected in 20 mosquito species, including ..."
[Response] Many thanks for your suggestion. We have modified it as suggested. (Page 10, lines 246-247)
- Figure 5, revise to "(A) Culex pipiens. (B) Aedes albopictus. (C) Culiseta longiareolata. (D) Ardea cinerea... etc. Inclusion of "vector of" and "host of" is redundant.
[Response] Many thanks for your suggestion. We have modified it as suggested. (Page 11, lines 292-293)
- Line 302. What are "risky" regions? Suggest revising to something more accurate.
[Response] Thanks for your valuable suggestion. We have revised this sentence as: “based on which the areas at potential risk for USUV infections were mapped, which covered the known distribution of USUV (Figure 6A), and identified additional regions at potential risk that were previously uninvestigated, mainly in Africa and Middle East (Figure 6B). The projected regions at potential risk for USUV infections (low-to-medium risk areas and high-risk areas) covered a potential suitable habitat of 1.80 million km2 inhabited by nearly 1.04 billion people.” (Page 12, lines 307-311)

Reviewer 3 Report
Comments and Suggestions for Authors
Chen J et al provide a comprehensive and exhaustive investigation of USUV infection based on publicly available data, demonstrating the numerous factors involved in the emergence of this virus. It is well and clearly written, well documented and well-illustrated although some figures could be improved.
Minor revisions:
Lines 33 and 88, Flavivirus should be replaced by Orthoflavivirus.
Line 39, please remove “the” USUV
Line 66, please change USUA by USUV
Line 247 there is an extra space between of and vectors
Table 1
· Legend, please correct by “human USUV infection” or “USUV infection in humans” or ‘human infection by USUV”.
· The % in the table should not be decimals as it is harder to read, rounding them would be relevant
· It is specified “Year” but it is more a “period” than a year that is specified.
· It would be relevant to add if patients were or not immunocompromised in this table
· Should be on pages 7 and 8/16
Figure 3
Is a very interesting figure, nonetheless, it is hard to determine the color of the point/polygon especially in figure 3A (between blue and green), 3B (between pink and red).
Globally this figure 3 as well as figures 4 and 5 should be larger for easier reading.
Figure 5
Moreover, the color legend is really hard to read.
Author Response
- Lines 33 and 88, Flavivirus should be replaced by Orthoflavivirus.
[Response] Thanks for your suggestion. According to your suggestion, we make modifications to this statement. (Page 1, line 33 & Page 2, line 92)
- Line 39, please remove “the” USUV.
[Response] We appreciate your suggestion. We have modified it as suggested. (Page 1, line 39)
- Line 66, please change USUA by USUV.
[Response] Many thanks for your suggestion. We have modified it as suggested. (Page 2, line 70)
- Line 247 there is an extra space between of and vectors.
[Response] Many thanks for your suggestion. We have modified it as suggested. (Page 10, line 253)
- Table 1
- Legend, please correct by “human USUV infection” or “USUV infection in humans” or ‘human infection by USUV”.
- The % in the table should not be decimals as it is harder to read, rounding them would be relevant.
- It is specified “Year” but it is more a “period” than a year that is specified.
- It would be relevant to add if patients were or not immunocompromised in this table
- Should be on pages 7 and 8/16
[Response] We appreciate your valuable suggestions. We have modified it as suggested. (Table 1 and Stable 8)
- Is a very interesting figure, nonetheless, it is hard to determine the color of the point/polygon especially in figure 3A (between blue and green), 3B (between pink and red).
Globally this figure 3 as well as figures 4 and 5 should be larger for easier reading
Figure 5 Moreover, the color legend is really hard to read.
[Response] Thank you so much for your helpful suggestions. Following your suggestion, we have adjusted the color for Figure 3, made the modifications to Figures 2, 3, 4, and 5 to improve their clarity, and redrawn Figure 3 to maintain the same background. (Figure 2, Figure 3, Figure 4, and Figure 5)

Round 2
Reviewer 2 Report
Comments and Suggestions for Authors
The authors have addressed all the reviewers comments.